# MoCtrl4D: Precise and Efficient Motion-Guided 4D Content Generation

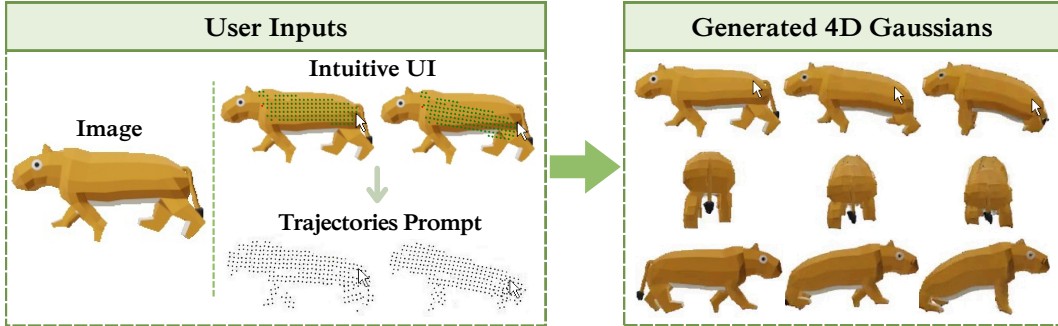

Figure 1: **MoCtrl4D.** Our method generates faithful 4D Gaussians given a single-view image and a motion prompt input from an intuitive user interface.

## Abstract

Promptable 4D generation is a crucial task with broad applicability across industries, thus has recently gained tremendous interest in research community. However, existing works remain predominantly limited to image and text conditioning, which neglect the nuances of motion controllability. To address this, we propose to use dynamic motion prompt defined by any number of point trajectories. To translate user intention into this motion representation, we design a user-friendly interface that allows users to intuitively input motion trajectories, bringing images to life through direct interaction. Unlike prior works, in leveraging prior knowledge of a base reconstruction model, our method integrates prompts without added modules, maintaining scalability and data efficiency without overhead, achieving a full forward pass in under a second. Furthermore, instead of relying on existing appearance-focused learning frameworks, which suffers from poor motion fidelity, we design a novel physically inspired *Vector Consistency Loss (VCL)* function for explicit motion learning. Our quantitative and qualitative results show significant improvement in spatiotemporally-precise and expressive control.

## 1 Introduction

The generation of high-quality 4D assets, defined as 3D models with dynamic temporal evolution, is a critical task with numerous applications. Its utility spans from creative industries such as 3D animation, film production, and game development to commercial and technical fields including augmented and virtual reality, product design, retail, and advertisement (Jiang et al. (2024); Xu et al. (2024); Zeng et al. (2024); Yu et al. (2024)). Traditional 3D animation is a labor-intensive process that requires skilled artists to model, rig, and animate assets frame by frame, which often takes over several days. Consequently, this domain has attracted widespread academic interest to automate the process.

One promising line of work is animatable 3D asset generation that automates the 3D modeling processes by automatic rigging (Xu et al. (2020); Baran & Popović (2007); Liu et al. (2025)) or parametric modeling (Liu et al. (2024); Pang et al. (2025); Cao et al. (2024); Taubner et al. (2025); Yu et al. (2025)). However, both approaches pose significant limitations on either degree of movement or types of objects. Most rigged-free and modeling-free 4D generation works are 4D reconstruction from 2D video (Wu et al. (2025a); Liang et al. (2024b); Wu et al. (2025b); Li et al. (2024); Chu

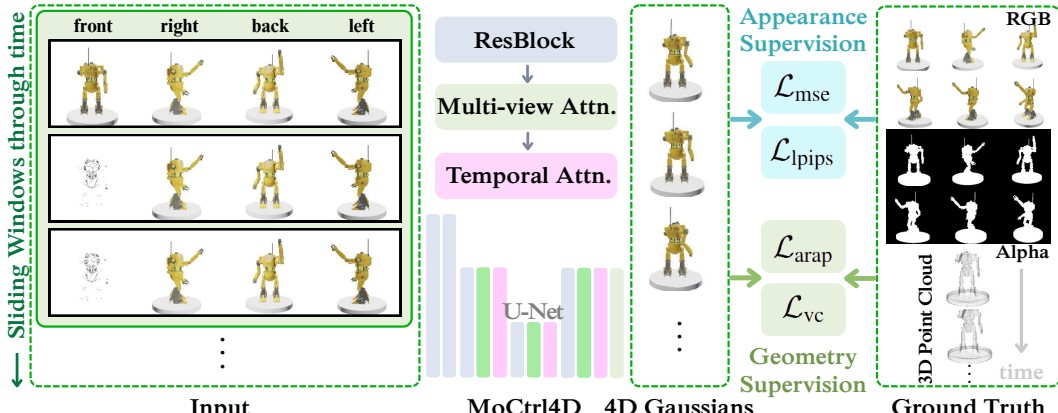

Figure 2: **Overall framework.** From the left, the input consists of four views. The main-view image (labeled "front") is formed by temporally concatenating the static image with the intended trajectory images. The auxiliary-view images are duplications of the initial-frame images across time. Built on a U-Net architecture, the model predicts 4D Gaussians through interleaved ResNet blocks enhanced by both multi-view and temporal attention layers. Training employs two supervisory signals: (1) appearance losses computed against ground-truth RGB and alpha renderings from the dataset, and (2) geometric motion losses computed against ground-truth 3D point trajectories.

et al. (2024); Wu et al. (2024c); Lei et al. (2025b); Huang et al. (2025); Xie et al. (2024); Wang et al. (2024)). These works effectively and faithfully lift sequences of 2D to 3D, eliminating the needs for skilled artists. However, they are limited by their reliance on existing video footage of the target motion. Recent advancements solve this by using text or image input to automatically generate 2D videos for the 4D reconstruction, introducing a new subfield: promptable 4D generation (Liang et al. (2024a); Yu et al. (2024); Jiang et al. (2024); Zeng et al. (2024)). These text/image-guided methods offer greater flexibility for users to generate 4D of desired. However, even though text prompts are expressive in high-level action and intuitive for users of any level of expertise, they remain innately ambiguous in 1) trajectories specification 2) movement speed or degree control of intended action, which cannot be solved by giving more detailed prompt. For instance, a text prompt like "move robot arm across the body" cannot control the precise path or trajectory in image space that a user may intend. These limitations are indicators that existing prompting methods remain incomplete in capturing full range of user intention. Therefore we aim to provide an alternative control mechanism to address this gap.

To this end, we present MoCtrl4D (Fig. 2), the first motion-conditioned 4D generation model that inputs single image and motion prompt to generate smooth and believable sequence of 3D Gaussians in feed-forward manner. Our approach employs sequence of arbitrary number of spatiotemporal positions of points or *point trajectories* (Geng et al. (2025), He et al. (2024) Sand & Teller (2008)) as our foundational representation. Point trajectories as a prompting signal stands out for 1) flexibility to any arbitrary number of points 2) precise spatiotemporal location, and 3) capability to represent any type of motion. Therefore, it captures the full spectrum of user intention. However, Generating 4D content from motion prompt requires a unified framework that simultaneously addresses several complex problems: (1) lifting a single 2D image into a coherent, temporally consistent 3D volume; (2) interpreting and integrating abstract, potentially sparse motion prompts in efficient way; and (3) synthesizing high-fidelity, physically plausible animations that are both view-consistent and faithful to the intention of users.

To address this critical challenges, inspired by Geng et al. (2025); Wu et al. (2024b); Li et al. (2025),we leverage prior knowledge from base video-to-4D reconstruction model to lift 2D to 3D. However, most existing methods process prompting signal by adding extra prompt-processing modules either by utilizing ControlNet or external module to encode the prompt (Zhang et al. (2023); Liang et al. (2024a); (Geng et al. (2025); Wu et al. (2024b); Li et al. (2025)), which suffer from additional parameters and computational overhead. We instead design motion-prompt injection mechanism that encode arbitrary number of point trajectories without requiring any modification or extension to the base model. Nevertheless, appearance-focused learning framework that yields promising results in our base 4D reconstruction model struggles to adapt to the new input signal. We therefore survey existing loss functions capable of providing explicit guidance for movement.

| Issue 1: False-Part Motion | | Issue 2: Fading Gaussians | |
|---|---|---|---|
| L4GM | MoCtrl4D | L4GM | MoCtrl4D |

Figure 3: **Limitation of L4GM framework on motion-prompted task.** Left: L4GM raises shoulder part instead of moving the robot arm ("1$^{st}$ View" denotes the prompted view). Right: when unable to converge on a plausible motion, L4GM optimizes by reducing Gaussian opacities to minimize the appearance losses.

The As-Rigid-As-Possible (ARAP) loss, as used in Lei et al. (2025b), Xiao et al. (2024), and Wu et al. (2025a), offers a valuable supervisory signal for learning collective motion without point-wise ground truth. However, a key limitation is its entanglement of rotational and translational components. This ambiguity arises because both movement modes maintain identical distance within a point pair, despite representing fundamentally different motions. To resolve this ambiguity and provide precise supervision, we introduce a novel *Vector Consistency Loss (VCL)*. Our ablation studies demonstrate that this loss function effectively complements the ARAP loss, leading to significantly more accurate movement learning. The contributions of this work are as follows:

- We are the first motion-conditioned 4D generation model. To the best of our knowledge, no prior method exists for nuanced motion-prompted 4D generation.
- Unlike prior motion-promptable 2D video generation methods, we are the first to inject motion prompt without adding extra prompt-processing modules to the base reconstruction model by utilizing the representation we term *trajectories image*.
- We invent the novel physically inspired *Vector Consistency Loss (VCL)* that effectively disambiguates ARAP loss and yields highly accurate per-Gaussian trajectory learning.

Qualitative and quantitative evaluations demonstrate the effectiveness of our method in expressive and believable 4D asset generation with an inference time of less than a second per forward pass.

## 2 RELATED WORK

**Motion-promptable 2D Video Generation.** Current state-of-the-art methods excel at generating high-quality 2D videos from a single image using motion control signals such as arrows keypoints or point trajectories (Geng et al. (2025); Wu et al. (2024b); Li et al. (2025); Zhang et al. (2025); Mou et al. (2024); Lei et al. (2025a); Burgert et al. (2025); Burgert et al. (2025); Tan et al. (2024)). Notably, Geng et al. (2025) demonstrates that point trajectories provide a particularly flexible and expressive prompt for fine-grained motion control. While these 2D video results are impressive, the application of such motion prompting for 4D assets remains critically unexplored. Our work bridges this gap by introducing an equivalently nuanced prompting mechanism for direct 4D content generation.

**Animatable 3D Object Generation.** Recent works automate the 3D modeling processes to enable subsequent smooth control by users for animation generation. One line of work facilitates the rigging process by automatic rigging (Xu et al. (2020); Baran & Popović (2007); Liu et al. (2025)). For example, RigAnything (Liu et al. (2025)) produces rig-ready 3D assets from image. This allows artists to bypass the time-consuming rigging process and proceed directly to the task of frame-by-frame animations, thereby 4D assets with user control can be made more efficiently. However, rigged-based approaches impose constraints on potential motion by the predefined degrees of freedom of the rig. Another line of work (Liu et al. (2024); Pang et al. (2025); Cao et al. (2024); Taubner et al. (2025); Yu et al. (2025)) employs parametric models to construct animatable 3D assets. These models utilize a fixed set of parameters to describe the pose and deformation of a character. While parametric models enable fine-grained detailed control, they are inherently specific to a narrow class of objects. This lack of generalizability renders them impractical for broader types of 4D assets.

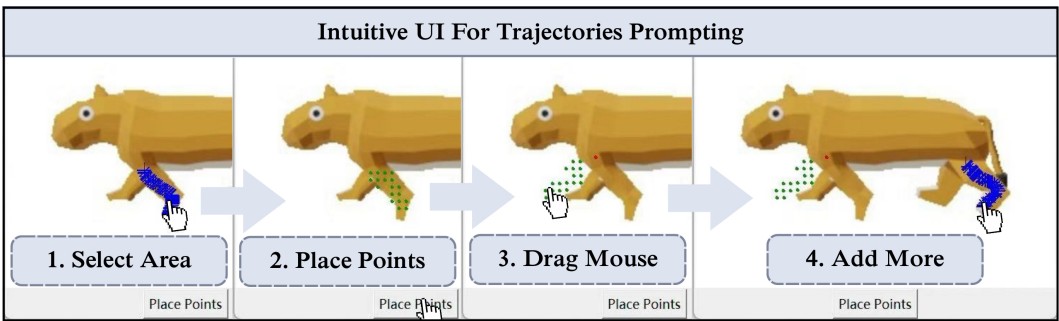

Figure 4: **Intuitive UI.** Users can easily give motion prompt to their desired image following these steps: 1. Mark intended area with blue brush. 2. Click "Place Points" button to automatically replace the area with uniformly distributed points 3. Choose one of the different modes of movement to dictate point trajectories via simple mouse dragging 4. Repeat step one to three again for new areas.

Considering these, we instead leverage rigged-free, and modeling-free approach for sequence of 3D generation to enable nuance controllability, and type-agnostic 4D generation.

**Promptable 4D Generation.** Recent approaches (Bahmani et al. (2024b); Ling et al. (2024); Jiang et al. (2024); Ren et al. (2023); Yu et al. (2024); Yin et al. (2023); Zheng et al. (2024); Liang et al. (2024a); Zeng et al. (2024)) use text/image as prompting methods. They mainly rely on a two-stage pipeline: text- or image-guided video generation via diffusion models, followed by an offline 4D reconstruction process (commonly using 4DGS Wu et al. (2024a)). TC4D (Bahmani et al. (2024a)) represents scene using a deformable 3D neural radiance field (NeRF) (Mildenhall et al. (2021); Li et al. (2022)) and utilizes control points to confine global trajectories of objects to move accordingly. A key limitation, however, is its inability to achieve fine-grained object-level control. While these approaches yield high-quality results, they suffer from two key limitations: 1) slow inference speed either due to the iterative nature of diffusion and optimization-based reconstruction or NERF representation, and 2) an inability to precisely control motion trajectories and degrees with coarse text/image prompts. We instead employ expressive and efficient motion prompting mechanism that complete the whole 4D generation pipeline in a single forward pass, trained end-to-end without additional module for prompt processing, enabling fine-grained control with high throughput.

**Feed-forward 4D Gaussian Reconstruction.** The iterative nature of optimization-based reconstruction makes it less preferable in applications where reconstruction speed is necessary. Tang et al. (2024), and Liang et al. (2024b) demonstrate the capability for feed-forward frameworks that can produce high-quality 3D or sequences of 3D content. They show that it is sufficient to utilize an appearance-focused learning framework, i.e., Mean Squared Error (MSE) and Learned Perceptual Image Patch Similarity (LPIPS) (Zhang et al. (2018)) losses to guide learning of the predicted Gaussian properties. This approach, though performs well in reconstruction, matches only appearance and neglects accurate gaussian motion. In contrast, our work employs motion-focused losses to facilitate learning of explicit motion.

## 3 METHOD

Given a static image and a motion prompt, our goal is to generate a 4D asset with believable and smooth motion that accurately follows the intended trajectories.

### 3.1 OVERALL FRAMEWORK

As illustrated in Fig. 2, given the initial object image and intended point trajectories at each time step $\mathbf{P} \in \mathbb{R}^{T \times N \times 2}$, where the $n^{th}$ track at the $t^{th}$ timestep is at coordinate $\boldsymbol{p}_t^n = (x_t^n, y_t^n)$, following Tang et al. (2024) and Ren et al. (2024), MoCtrl4D adopt two-step pipeline.

In the first step, at inference, we utilize an existing multi-view image diffusion model to obtain four images at predefined orthogonal azimuths at fixed elevation. To handle point trajectories, which vary in number, we first encode them into a standardized format we term *trajectory images*, $\mathbf{I}_{tr} \in$

$\mathbb{R}^{T \times H \times W \times 3}$ (detailed in Sec. 3.2). This representation is subsequently concatenated temporally with the initial four-view images. Note that we only take trajectory prompt in the main view for practical user input consideration. For other supporting views, we simply duplicate the initial static images across time:

$$\mathbf{C}^v = \begin{cases} \mathbf{I}_{img}^v \oplus \mathbf{I}_{tr}^v & \text{if } v = 0, \\ \mathbf{I}_{img}^v & \text{otherwise} , \end{cases} \quad (1)$$

where $\mathbf{C}$, $\mathbf{I}_{img}$, $\oplus$, and $v$ represent input RGB tensor, static object image, temporal concatenation operation, and $v^{th}$ view respectively. $v{=}0$ denotes the main input view. Subsequently, these RGB tensors are concatenated channel-wise with Plücker ray embedding (Xu et al. (2023)). The input is therefore:

$$\boldsymbol{f}_i = \{\boldsymbol{c}_i, \boldsymbol{o}_i \times \boldsymbol{d}_i, \boldsymbol{d}_i\}, \quad (2)$$

where $f_i, c_i, o_i,$, and $d_i$ represent input feature, RGB value, ray origin, and ray direction for pixel $i$ respectively. In the second step, we utilize L4GM (Ren et al. (2024)), a U-Net-based video-to-4D reconstruction model with spatial and temporal attention layers, as our base model. Moctrl4D is trained to instead take images with motion prompt to lift them into sequences of corresponding 4D Gaussians.

If the motion prompt contains more frames than the input window of our model, we autoregressively process sequences of object and trajectory images using a sliding temporal window. For the initial window, the input image $\mathbf{I}_{img}$ is provided directly by user. For all subsequent windows, $\mathbf{I}_{img}$ is generated by rendering the predicted 3D Gaussians from the last frame of the previous time step into the four predefined orthogonal views. This rendered output is then concatenated with the remaining trajectory images to form the input for the next prediction step.

## 3.2 Motion Prompting without Extra Modules

Motion-promptable video generation methods (Geng et al. (2025); Wu et al. (2024b); Li et al. (2025)) typically build upon a base video generation model and rely on ControlNet (Zhang et al. (2023)) or external module to incorporate motion prompts. However, this approach duplicates parts of the network, leading to significant increases in both computational and time complexity.

Motivated by this, we invent a novel motion-prompt injection method that requires no additional modules nor architectural modifications to the base model. To elaborate, we convert motion trajectories into corresponding *trajectory images* $\mathbf{I}_{tr} \in \mathbb{R}^{T \times H \times W \times 3}$.

Specifically, inspired by MotionPrompting Geng et al. (2025), we first take user-specified motion prompting in the form of point coordinates at each time step $\mathbf{P} \in \mathbb{R}^{T \times N \times 2}$. We subsequently assign randomly sampled unique RGB values to each track. Subsequently, for each spatiotemporal coordinate $\boldsymbol{p}_t^n = (x_t^n, y_t^n)$ of the $n^{th}$ trajectory, we add its unique RGB value to the initially blank point trajectory images. In cases where multiple trajectories overlap, their embeddings are additively blended. This prompt representation possesses two crucial properties: 1) Direct compatibility: they serve as a native input to the existing base model, avoiding the need for a separate control module, and 2) Density-invariance: they are independent of number of points as the input trajectories are converted into a tensor of a fixed, predefined shape regardless of the number of tracks. Consequently, the system is capable of handling motion control signals with any arbitrary number of point trajectories ranging from very sparse to completely dense.

## 3.3 Motion Learning

While L4GM (Ren et al. (2024)) framework, utilizing only appearance losses namely Mean Squared Error (MSE) and Learned Perceptual Image Patch Similarity (LPIPS) (Zhang et al. (2018)) losses, effectively reconstructs sequences of 3D Gaussians, its inherent limitation in modeling dynamic geometry hinders performance on motion-prompted tasks: model frequently fails to learn valid motions for challenging prompts, opting instead for degenerate solutions that cheat the appearance losses. As illustrated in Fig. 3, we found two main issues: 1) False-Part Motion: When a motion prompt intended for one object part results in the implausible movement of a different part. An example case is prompting a robot arm to move across the body: the expected arm motion is replaced by an implausible elevation of the shoulder. 2) Fading Gaussians: model is optimized to reduce

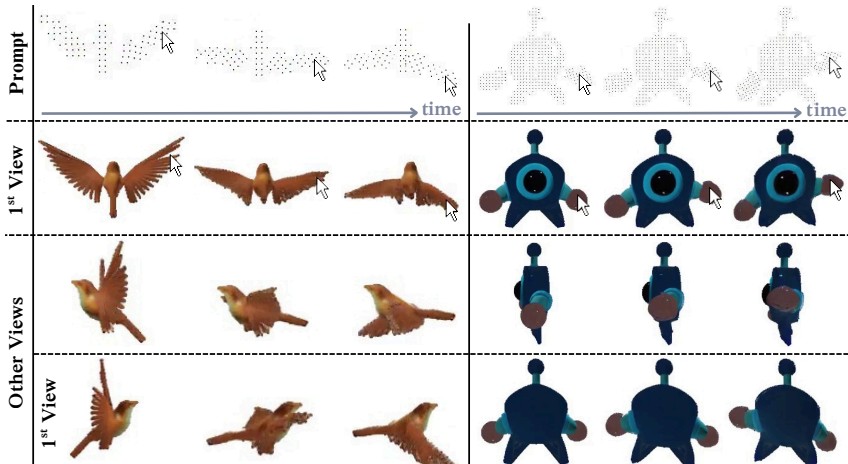

Figure 5: **User-prompted 4D Generation.** Our method generates physically plausible and natural motion from user prompts.

Gaussian opacities to minimize appearance losses when unable to converge on a physically plausible motion. This usually happens when prompted with sparse trajectories. Note that this tendency to reduce opacity persists even with samples that L4GM correctly predict movement, resulting in texture inconsistency. Firstly, to prevent fading, we restrict the model to predict the full set of Gaussian properties, which are mean position $\chi$, scale $s$, rotation $r$, opacity $\alpha$, and color $c$, solely at the initial frame $G_0 = \{\chi_0, s_0, r_0, \alpha_0, c_0\}$. For subsequent frames, the model predicts only the selected geometric properties, specifically position and rotations $G_t = \{\chi_t, s_0, r_t, \alpha_0, c_0\}$.

Secondly, we examine possible objective functions to provide movement supervision to the model. We ideally need ground-truth positions for each predicted 3D Gaussian at every timestep during training, which is not feasible. Hence, we build on the observation from Lei et al. (2025b) and Xiao et al. (2024) that most real-world deformable motion can be decomposed into subparts that behave as if rigidly attached. Leveraging this insight, we constrain the motion of 3D Gaussians within the same subpart to move rigidly together over time. Specifically, we achieve this by employing a physics-inspired As-Rigid-As-Possible (ARAP) loss:

$$\mathcal{L}_{\text{arap}} = \frac{1}{T|\Omega_{i,j}|} \sum_{t=1}^{T} \sum_{\Omega_{i,j}} s_{i,j} \Big| \left|\left|\boldsymbol{\chi}_{i,t} - \boldsymbol{\chi}_{j,t}\right|\right|_2 - \left|\left|\boldsymbol{\chi}_{i,0} - \boldsymbol{\chi}_{j,0}\right|\right|_2 \Big|, \tag{3}$$

where $T, \Omega_{i,j}$, and $s_{i,j}$, represent the number of input frames, the collection of all pairwise indices, and affinity score of rigidity of $i, j$ pair respectively.

However, because the ARAP loss only preserves the relative distance between point pairs, it introduces ambiguity. This permits degenerate solutions: for instance, a Gaussian pair that should undergo pure translation might instead rotate together while still minimizing the loss, rendering this constraint less effective for motion learning. To eliminate this ambiguity, we carefully design a novel loss function that clarifies exactly one mode of movement: translation. Specifically, we propose *Vector Consistency Loss (VCL)*, defined as

$$\mathcal{L}_{\text{vc}} = \frac{1}{T|\Omega_{i,j}|} \sum_{t=1}^{T} \sum_{\Omega_{i,j}} \sigma_{i,j} \Big|\Big|(\boldsymbol{\chi}_{i,t} - \boldsymbol{\chi}_{j,t}) - (\boldsymbol{\chi}_{i,0} - \boldsymbol{\chi}_{j,0})\Big|\Big|_1, \tag{4}$$

where $\sigma_{i,j}$ represents affinity score of vector consistency of $i, j$ pair. Unlike ARAP loss that regulates scalar distance, VCL constrains the relative position vector between a pair of Gaussian means. So, when combined, ARAP ensures rigidity within rigid sub-parts, while VCL sharpens the constraint for special subsets.

We derive $s_{i,j}$ and $\sigma_{i,j}$ from ground truth 3D point cloud trajectories. Specifically, we first compute an affinity score for every pair of points based on their motion. Then, for each predicted Gaussian, we find its nearest neighbor in the ground truth point cloud and assign the corresponding

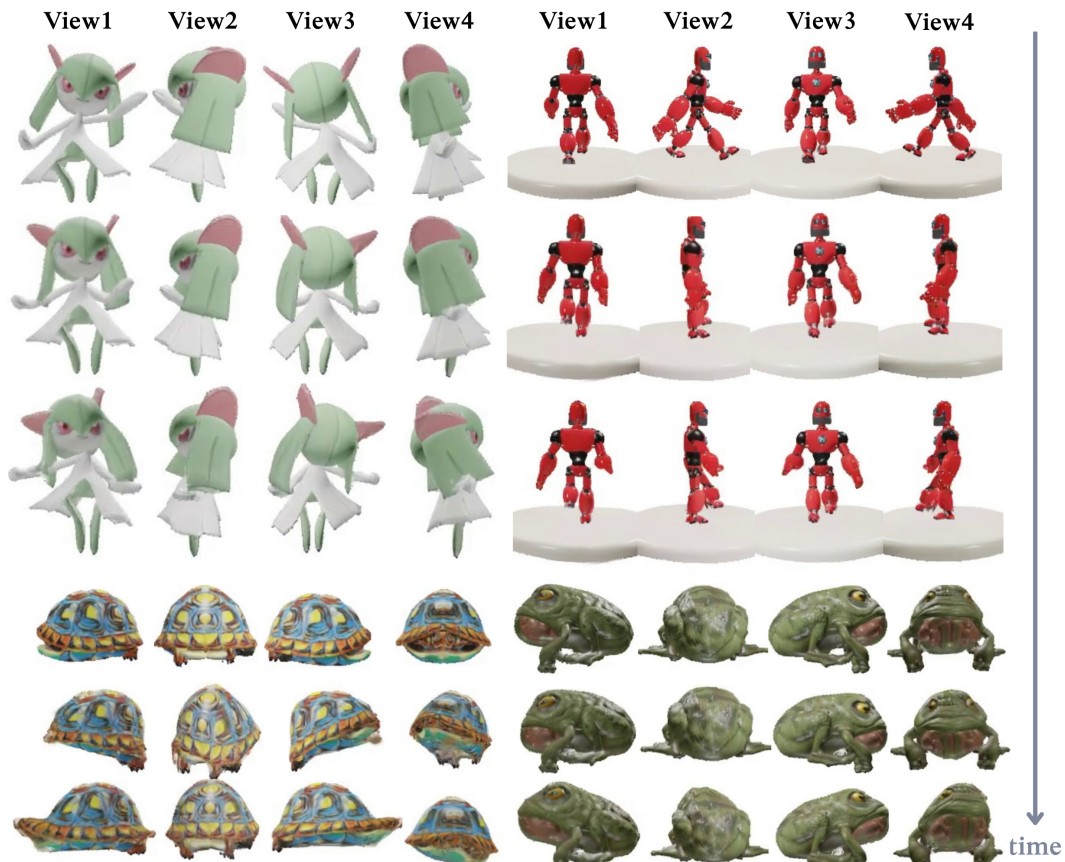

Figure 6: **Main results.** Our method demonstrates expressive motion expression and consistent high-quality texture.

pre-computed affinity scores. Specifically, for affinity score of rigidity:

$$s_{i,j} = \begin{cases} 1 & \text{if } \sum_{t=1}^{T} \left| \left\| \boldsymbol{\chi}_{i,t} - \boldsymbol{\chi}_{j,t} \right\|_2 - \left\| \boldsymbol{\chi}_{i,0} - \boldsymbol{\chi}_{j,0} \right\|_2 \right| < \tau_1, \\ 0 & \text{otherwise,} \end{cases} \tag{5}$$

and affinity score of vector consistency:

$$\sigma_{i,j} = \begin{cases} 1 & \text{if } \sum_{t=1}^{T} \left\| (\boldsymbol{\chi}_{i,t} - \boldsymbol{\chi}_{j,t}) - (\boldsymbol{\chi}_{i,0} - \boldsymbol{\chi}_{j,0}) \right\|_1 < \tau_2, \\ 0 & \text{otherwise,} \end{cases} \tag{6}$$

where $\tau_1$ and $\tau_2$ denote threshold values.

We jointly optimize the original appearance losses and the two geometry constraints. The combined loss function is therefore:

$$\mathcal{L} = \lambda_{\text{mse}} \mathcal{L}_{\text{mse}} + \lambda_{\text{lpips}} \mathcal{L}_{\text{lpips}} + \lambda_{\text{arap}} \mathcal{L}_{\text{arap}} + \lambda_{\text{vc}} \mathcal{L}_{\text{vc}}. \tag{7}$$

## 4 EXPERIMENTS

### 4.1 DATASETS

To train our model, multi-view videos and ground-truth 2D point trajectories are required. This condition harshly limits the selection of suitable datasets, as there is no available dataset that readily satisfies these conditions. The Objaverse dataset (Deitke et al. (2023)), which provides 3D animations, aligns most closely with our requirements. Specifically, we use a subset of Objaverse curated by Jiang et al. (2024).

Figure 7: **Ablation.** Qualitative comparison between each ablation experiment denoted as 1 to 5, corresponding to each line in Tab. 4. "Gt." denotes ground truth images.

**Image Rendering.** Inspired by Wang & Shi (2023), Melas-Kyriazi et al. (2024), Li et al. (2023), and Ren et al. (2024), we render each 4D object from 24 distinct viewpoints. These comprise: 1) 8 fixed cameras positioned at zero degrees elevation with uniformly increasing azimuth, and 2) 16 cameras with randomized positions. We then discard almost all white rendered images. This results in 2M frames of animations from 10K animated 3D objects.

**2D Point Trajectories Extraction.** The dataset represents dynamic 3D objects as static meshes, with animations encoded as bone transformations of the rigged models. To derive ground-truth 2D point trajectories from this dataset, we first extract the point coordinates of the 3D object in its first frame. Subsequently, we derive the next-frame coordinates by calculating Linear Blend Skinning (LBS) (Sumner et al. (2007)) as follows:

$$\boldsymbol{v}' = \sum_{i=0}^{N_1}(W_i \times \boldsymbol{M}_i \times \boldsymbol{v}) \,, \tag{8}$$

where $\boldsymbol{v}', N, W_i, \boldsymbol{M}_i$, and $\boldsymbol{v}$ represents the deformed coordinate of the vertex, the total number of bones influencing the vertex, skinning weight for the influence of the i-th bone on the vertex, transformation matrix for the i-th bone, and previous vertex coordinate. This approach allows us to obtain complete 3D point trajectories that accurately capture motion of the object throughout the sequence. Since we use EEVEE rendering engine of Blender (Foundation (2019)) to render RGBA sequences, we retrieve camera parameters, including camera intrinsics and extrinsics, to construct the projection matrix.

### 4.2 IMPLEMENTATION DETAILS

**Training.** We use 8 supervision views; 4 from input views and 4 from the remaining pool of rendered views are used. We split each rendered animation into one-second subclips. We set T=3 and stride each sub-clip by 3 to prevent near-static animations training. We initialize the model with L4GM weights while keeping the downsampling blocks frozen for training data efficiency. Following Tang et al. (2024) and Ren et al. (2024), We adopt AdamW (Loshchilov & Hutter (2017)) with learning rate of $4^{-4}$. We train our model on 8 NVIDIA GeForce RTX 3090 (24G) GPUs with effective batchsize of 48 for 30 epochs. We also adopt the augmentation strategy from LGM (Tang et al. (2024)), which includes grid distortion and orbital camera jitter. For motion losses, we sample 2,000 Gaussians each iteration and use affinity score threshold of $1 \times 10^{-8}$ for both ARAP and VCL. The loss weights are set as follows: $\lambda_{\text{mse}} = \lambda_{\text{lpips}} = \lambda_{\text{vc}} = 1$ and $\lambda_{\text{arap}} = 1000$.

**Inference.** To enable translation of users intention to trajectory prompt, we design a simple intuitive user interface (UI) that enables users to define motion trajectories through painting and dragging actions. As illustrated in Fig. 4, the workflow is as follows: users first specify intended duration to input motion prompt. Subsequently, they can paint over the target areas with a blue brush to select regions for points placement. After clicking "Place Points," the system automatically replaces these markings with a uniform set of green points, positioned exclusively on the objects. Users can then choose between different movement modes such as moving points along a circular arc about a pivot or directly dragging these points freely as a group. This process can be repeated for multiple object groups: users may paint a new area to place and animate additional points.

Table 1: **Quantitative Evaluations.** We evaluate the appearance (PSNR, SSIM, LPIPS) and movement (EPE) metrics on the testing subset of the Objaverse dataset. Random denotes uniform random from 500 to 1000 number of tracks (inclusive).

| # Tracks | Method | PSNR ↑ | SSIM ↑ | LPIPS ↓ | EPE ↓ |
|---|---|---|---|---|---|
| N = 1 | L4GM | 22.532 | 0.911 | 0.073 | **0.196** |
| | MoCtrl4D | **22.841** | **0.914** | **0.071** | 0.205 |
| N = 16 | L4GM | 23.161 | 0.917 | 0.067 | 0.203 |
| | MoCtrl4D | **23.381** | **0.919** | **0.065** | **0.169** |
| N = 512 | L4GM | 24.168 | 0.926 | 0.057 | 0.163 |
| | MoCtrl4D | **24.364** | **0.928** | **0.056** | **0.131** |
| N = 2048 | L4GM | 24.278 | 0.928 | 0.056 | 0.146 |
| | MoCtrl4D | **24.467** | **0.929** | **0.055** | **0.131** |
| Random | L4GM | 24.198 | 0.927 | 0.057 | 0.162 |
| | MoCtrl4D | **24.393** | **0.929** | **0.056** | **0.129** |

## 4.3 EVALUATIONS

**Quantitative Evaluation.** We utilize Objaverse testing subset for evaluation as shown in Tab. 1

**Qualitative Evaluation.** Fig. 5 illustrates the resulting 4D generation from user prompts. MoCtrl4D generates high-quality 4D sequences that precisely follow the intended motion trajectories. Fig. 6 shows the resulting 4D generation on Objaverse test set.

**Evaluation Metrics.** We use testing subset of Objaverse 1.0. For appearance evaluation, we compute average PSNR, SSIM (Wang et al. (2004)), and LPIPS (Zhang et al. (2018)) between the generated 4D rendered into the supervision views and ground truth of the same views. To evaluate trajectories accuracy, we calculate average 3D end-point error (EPE), defined by L2 distance between ground truth points and gaussian means if their initial positions are approximately at the same positions. Since our predicted gaussians are confined in a [-1,1] range, we scale EPE by 10 for readability. Note that all evaluation is done in autoregressive setting for 3 windows.

**Ablation.** Model and training designs are ablated as shown in Tab. 4, experiments 3 and 5 show that preserving appearance yields superior performance. Experiments 4 and 5 confirm the effectiveness of the Vector Consistency Loss (VCL). Note that we use fewer training steps for ablation studies. We also show qualitative comparisons of ablation study in Fig 7.

We ablate the output time duration as presented in Tab. 4.3. At equivalent output durations, using more input frames during training yields improved output quality. While increasing the number of inference frames and autoregressive windows enables handling longer motion prompts, this comes with a corresponding drop in performance. Collectively, these results demonstrate that video duration can be extended while incurring only modest degradation in output quality.

Table 2: **User study.** User study evaluating the quality of generated 4D assets for motion-promptable image-to-4D tasks. Ratings are on a 1–5 scale, with higher scores indicating better quality. Videos used to conduct the study are available on project page (ablation section).

| | Video Consistency ↑ | Overall Quality ↑ |
|---|---|---|
| L4GM | 3.11 | 2.97 |
| MoCtrl4D | **3.92** | **3.04** |

**User Study.** We have conducted user study on 14 prediction results between L4GM and MoCtrl4D as shown in Tab. 2

## 4.4 LIMITATIONS

Although the results are promising, our method remains subject to certain limitations. 1) MoCtrl4D is currently specialized for artwork-style images. Because no readily available dataset of real-world

Table 3: **Comparison of Video Length Extension Methods.** This table evaluates how the number of input frames (during training and inference) and autoregressive windows impact output quality metrics (PSNR, LPIPS). Results show that video duration can be extended while incurring minor performance degradation. Note that the models in this table were trained for fewer epochs than the training duration used for our main results.

| #Input frames at training | #input frames at inference | #Autoregressive windows | #Total output frames | PSNR | LPIPS |
|---|---|---|---|---|---|
| 3 | 3 | 1 | 3 | 24.9115 | 0.0527 |
| | | 2 | 6 | 24.817 | 0.052 |
| | | 3 | 9 | 24.060 | 0.060 |
| | | 4 | 12 | 23.492 | 0.065 |
| | 8 | 1 | 8 | 23.730 | 0.058 |
| | | 2 | 16 | 23.074 | 0.071 |
| | | 3 | 24 | 23.007 | 0.073 |
| | | 4 | 32 | 22.721 | 0.078 |
| 8 | 8 | 1 | 8 | 24.867 | 0.053 |
| | | 2 | 16 | 24.113 | 0.062 |
| | | 3 | 24 | 23.747 | 0.067 |
| | | 4 | 32 | 23.475 | 0.071 |
| | 16 | 1 | 16 | 24.220 | 0.060 |
| | | 2 | 32 | 23.782 | 0.067 |
| | | 3 | 48 | 23.560 | 0.071 |
| | | 4 | 64 | 23.381 | 0.075 |

Table 4: **Ablation.** We ablate the training methods. Result shows that ours (experiment 5) works best. Note that all methods are tested with random number of tracks from 500 to 1000 (inclusive) with the same random seed.

| Predict appearance | # | Method | PSNR ↑ | SSIM ↑ | LPIPS ↓ | EPE ↓ |
|---|---|---|---|---|---|---|
| All frames | 1 | No finetune (L4GM weights) | 15.946 | 0.831 | 0.240 | 0.330 |
| | 2 | L4GM framework | 21.101 | 0.872 | 0.101 | 0.190 |
| | 3 | w/ ARAP+VCL | 21.587 | 0.875 | 0.097 | 0.178 |
| Initial frame | 4 | w/ ARAP | 21.154 | 0.884 | 0.094 | 0.164 |
| | 5 | w/ ARAP+VCL | **21.622** | **0.897** | **0.077** | **0.156** |

4D assets exists, our model is trained only on objects in such artistic domains, making real-world photographs out-of-distribution. Achieving robust generalization to in-the-wild images will require a carefully designed self-supervised framework that can leverage in-the-wild data during training. 2) Although many real-world deformations can be decomposed into rigid subparts (Lei et al. (2025b); Xiao et al. (2024)), highly elastic or fluid-like cases fall outside this assumption. VCL focuses on local continuity to rigid-subpart consistency and does not model such higher-order physical dynamics.

## 5 CONCLUSION

This work presents the first framework for motion-prompted 4D generation. Unlike prior methods that rely on extra modules, our approach injects prompts via a novel *trajectory image* natively compatible with the reconstruction model. We explicitly train the model to learn accurate per-Gaussian motion by constraining excessive degrees of freedom and adding two key losses: a physically inspired ARAP loss, complemented by our novel *Vector Consistency Loss (VCL)*, which disambiguates movement modes within rigid subgroups. Our approach achieve both high-fidelity and believable motion and appearance in 4D generation. Finally, a simple-to-use UI demonstrates the versatility and practical applicability of our method.

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
