# OpenReview forum: "MoCtrl4D: Precise and Efficient Motion-Guided 4D Content Generation"
_ICLR.cc/2026/Conference — Submitted to ICLR 2026_

### Official Review · Reviewer_RLzK · 2025-10-16

**Soundness:** 2
**Presentation:** 2
**Contribution:** 2
**Rating:** 2
**Confidence:** 5

**Summary:**

This paper proposes MoCtrl4D, a pipeline for controllable 4D generation. Compared to previous works, which mainly rely on text to define motion control, this work controls motion through point trajectories. Given static images and motion trajectories, the method synthesizes 4D Gaussians that can be rendered from any viewpoint and timestep. The model is supervised with appearance and geometry losses separately.

**Strengths:**

- Important task: Motion-controllable 4D generation is an important and underexplored task.

**Weaknesses:**

- Weak results: The quantitative results are not very convincing, and the qualitative results lack much motion. The videos are generally super short and it is not clear why the motion is so limited, since Objaverse training data does have larger amount of motion.
- Weak result presentation: The supplementary video material, crucial for such 4D generation papers, is not very polished. I highly recommend a supplementary website with side-by-side comparisons with previous works. The results also do not have any full 4D visualizations, i.e., space-time renderings where both time and viewpoint are changing, as well as freeze-time and freeze-camera renderings.
- Missing comparisons: The paper does not compare with motion-controllable 4D generation methods, e.g., MagicPose4D [1] or SP4D [2]. It mainly compares with L4GM, a feed-forward 4D reconstruction method that models each time separately and does not have any motion control.

[1] Zhang et al., MagicPose4D: Crafting Articulated Models with Appearance and Motion Control, TMLR 2025 \
[2] Zhang et al., Stable Part Diffusion 4D: Multi-View RGB and Kinematic Parts Video Generation, NeurIPS 2025

**Questions:**

I am convinced by the task the paper is trying to solve but the results are not convincing.
I would like authors to address following questions:
- Why is the motion is so limited?
- How does the method compare to MagicPose4D or SP4D, i.e., other motion-controllable 4D generation frameworks?

I am pretty negative and it is rather unlikely that I will raise my score to an accept but I am still happy to consider a rebuttal.

---

> ### Author Response · Authors · 2025-12-03
>
> We deeply value the reviewer’s time, expertise, and thoughtful comments alongside constructive questions. Your input strengthens our work, and we offer our detailed clarifications here:
>
>
> $\textbf{Q1: Why is the motion so limited?}$
>
> Please note that the motion is not actually limited. We apologize that in the initial supplementary materials we randomly selected samples from the test set, many of which had small amplitudes, leading to insufficient demonstration and misunderstanding regarding the limits of motion generation. We have provided updated visualizations on our project page that clearly demonstrate the high amplitude and variability of our motion generation.
> Project Page: https://moctrl4d.github.io/moctrl4d/
>
>
> $\textbf{Q2: How does the method compare to MagicPose4D or SP4D, i.e., other 4D generation frameworks?}$
>
> Since MagicPose4D isn’t currently providing full implementation code, we cannot fully compare with it. However, we do provide comparison with LASR, 4D reconstruction model recommended in the pipeline of MagicPose4D. We also show comparison with SP4D on our project page. MoCtrl4D significantly outperforms both methods.

---

### Official Review · Reviewer_Nryf · 2025-10-19

**Soundness:** 2
**Presentation:** 1
**Contribution:** 3
**Rating:** 2
**Confidence:** 4

**Summary:**

This paper introduces MoCtrl4D, a feed‑forward, motion‑prompted 4D content generation framework that lifts a single image plus user‑drawn point‑trajectory prompts into dynamic 3D Gaussians with an intuitive UI and sub‑second forward pass claims. The core ideas are a trajectory‑image injection that avoids extra control modules and a Vector Consistency Loss (VCL) that complements ARAP to supervise per‑Gaussian motion more precisely than appearance‑only training.

**Strengths:**

1. Introduces a direct motion‑prompt mechanism using point trajectories that provides fine‑grained, expressive control beyond text/image prompts.​
2. Proposes “trajectory images” that integrate seamlessly into a base reconstruction model, avoiding ControlNet‑style module duplication and extra parameters.
3. Novel VCL loss for more specialised motion control.
4. Mitigates opacity‑driven degeneracies by predicting appearance only at the first frame while updating geometry over time, improving temporal consistency.

**Weaknesses:**

1. The presentation is extremely poor. There is no direction to understand the supplementary material videos. I suggest using a html page so that the reviewers understand what the videos indicate and what exactly to interpret. Authors have just poured in a bunch of videos in the suplementary material without proper direction to understand them.

2. As a work on dynamics and 4D content generation, I would request atleast seeing 20-30 videos to see the fidelity of the work and not just assume that these are just cherry-picked results. Also few videos in the supplementary almost have no motion at all. It would also be good to see the same object undergoing different motion trajectories, like atleast 2-3 different motions to understand the generalisability of the model. So same object but different motion points to get different 4D gaussians and visualise their dynamics.

3. The paper is also really poorly written. It is extremely difficult for the reviewer to understand what the author is trying to convey at several places like: Fig 2 caption: Figure caption should be extremely descriptive and should describe each module properly. In the figure, you have kept a bar graph of what I assume is Resblock, Multi-view attn and Temporal attn but what do the height signify? Please be more clear. Also in the input, I am really confused as to what exactly is the input? Why does the first row contain 4 images and then 2nd and 3rd row to have one point image and 3 rgb images and again 4th row contains different set of rgb images. Please properly describe the input and output space.

4. The comparisons are very limited. I don’t see a single image except Fig 3, which has only comparison with one method on only 2 data samples. I would request the authors to provide more comparisons on more scenes and also comparisons with more baselines and previous works. This level of comparison is not acceptable for the standard of this conference.

5. It would also be great to include a user-study done on atleast 10-15 scenes with varying object motions and view points. More photos of the UI could be shown atleast in the appendix to understand the flexibility of motion control in the UI.

6. How does the model scale to real-world objects. All the results are shown on synthetic dataset which is good but what matters is results on real-world examples. It would be great if you could show results on few real-world objects.

7. Also how does the model scale to longer motion trajectories. There are no discussions on failure cases which I recommend adding and explaining the limitations of the work.

**Questions:**

I have highlighted the major questions in the weaknesses already, and I am summing them up below(I have summarised them shortly so that it is easier for the reviewer to quote the exact question they are answering. Please refer to the Weakness for the detailed problem and question asked.

1. Could you organize the supplementary videos in a clearer way, perhaps using an HTML page, so that it’s easier for readers to understand what each video shows and what we’re supposed to interpret from them? Right now, the supplementary material feels like a random collection of videos without structure or explanation.

2. Why are there so few videos presented? Since this paper focuses on 4D dynamics and content generation, it would be good to see at least 20–30 videos to properly judge fidelity. With so few examples, it’s hard to tell if the results are representative or cherry-picked.

3. Some supplementary videos seem to have almost no motion at all. Could you explain why that is the case? Are those examples meant to show static scenes, or is the model struggling to generate realistic motion?

4. Could you include examples where the same object undergoes different motion trajectories, say two or three variations, to test whether the model generalizes well and to better visualize its learned 4D Gaussian dynamics?

5. In Fig. 2, could you clarify what the bar heights represent for the ResBlock, Multi-View Attention, and Temporal Attention modules? The caption and figure are not clear on what those heights signify.

6. What exactly is the input to the model? The figure is confusing because the first row shows four images, the second and third rows have one point image and three RGB images, and the fourth row has a completely different combination. Could you describe the input and output spaces more clearly?

7. Why are the comparisons so limited? Apart from Fig. 3, which only compares to one method on two samples, there are no other comparisons. Could you add more comparisons with other baselines and across a wider range of scenes?

8. Would it be possible to include a small user study on around 10–15 scenes with varying object motions and camera viewpoints? That would help quantify perceptual quality and realism.

9. Could you include more images of the UI (perhaps in the appendix) to show how flexible the motion control interface is?

10. How does the method perform on real-world scenes? Currently, all the results seem to be on synthetic data. It would strengthen the paper to show at least a few examples on real-world objects.

11. How does the method handle longer motion trajectories? There is no discussion of whether performance degrades or artifacts appear with extended motion.

12. Could you include a section discussing failure cases and the main limitations of your approach? This would help readers understand where the method works well and where it struggles.

---

> ### Author Response · Authors · 2025-12-03
>
> We are truly grateful to the reviewer for the incredibly thoughtful, detailed comments and constructive questions. Your insights have been invaluable in helping us refine our work and strengthen our submission. Please find our detailed clarifications in response to your feedback below:
>
>
> $\textbf{Q1: Could you organize the supplementary videos more clearly—perhaps with an HTML page?}$
>
> We apologize for the unclear presentation of the initial supplementary material. Here is the organized project page: https://moctrl4d.github.io/moctrl4d/
>
>
> $\textbf{Q2: Request for 20-30 videos.}$
>
> We apologize for previous small number of videos in the supplementary material. We present 30+ videos including failure cases in the project page.
>
>
> $\textbf{Q3: Some supplementary videos seem to have almost no motion at all. Why?}$
>
> Almost-static cases are from testing set. We show them as representative of model performance on the test set. We present larger movement on our project page.
>
>
> $\textbf{Q4: Request for same object undergoing different motion trajectories.}$
>
> We thank reviewer for the valuable suggestion. MoCtrl4D generalizes well to different motion trajectories. We present the motion variety on the project page.
>
>
> $\textbf{Q5: In Fig. 2, could you clarify what the bar heights represent for the ResBlock, Multi-View Attention,}$
> $\textbf{and Temporal Attention modules?}$
>
> We apologize for the vagueness. Bar heights of Fig. 2 illustrate asymmetric U-Net [1] architecture of the base model L4GM [2]. The height of each bar represents the different sizes of spatial resolutions. This model architecture is the interleave of residual layers [3], Multi-View Attention, and Temporal Attention modules. We have improved clarity of the description in the edited version.
> [1] U-Net: Convolutional Networks for Biomedical Image Segmentation
> [2] L4GM: Large 4D Gaussian Reconstruction Model
> [3] Deep residual learning for image recognition.
>
>
> $\textbf{Q6: What exactly is the input to the model? The figure is confusing.}$
>
> We apologize for unclear figure and its description. We have edited them for better clarity and we explain here more details about the input: as written in eq. 1, if v=0 (main view, which is the leftmost of Fig.2), the input is the temporal concatenation of the initial object image and trajectory images, otherwise (all other three views, which are the second to the last column of Fig.2), the visual input is duplication of the initial static images across time for each window.
>
>
> $\textbf{Q7: Comparisons with other baselines and across a wider range of scenes.}$
>
> We added comparison with LASR and SP4D in our project page including humanoid, monster, and object type. MoCtrl4D significantly outperforms both methods.
>
>
> $\textbf{Q8: A small user study on around 10–15 scenes.}$
>
> ### Table 1: User study evaluating the quality of generated 4D assets for motion-promptable image-to-4D tasks. Ratings are on a 1–5 scale, with higher scores indicating better quality. Videos used to conduct the study are available on project page (ablation section).
> |          | Video Consistency ↑ | Overall Quality ↑ |
> |----------|---------------------|-------------------|
> | L4GM     | 3.11                | 2.97              |
> | MoCuR4D  | 3.92                | 3.04              |
>
>
> $\textbf{Q9: Show how flexible the motion control interface is}$
>
> We provide a demonstration video on the $\textit{project page}$ showcasing our user-friendly interface. The video highlights that our method eliminates the need for key-frame manipulation, in contrast to traditional 3D animation frameworks, which depend on highly skilled artists for key-frame animating.
>
>
> $\textbf{Q10: How does the method perform on real-world scenes?}$
>
> MoCtrl4D currently specializes for artwork-style images. Since there is no readily available dataset for real world 4D assets. To be able to generalize to in-the-wild images requires careful self-supervised framework to train the model using in-the-wild data. We thank reviewer for the advice for further work.
>
>
> $\textbf{Q11: How does the method handle longer motion trajectories? discussion of whether performance degrades}$
> $\textbf{or artifacts appear with extended motion}$
>
> There are two methods to handle longer motion trajectories. One is autoregression and another is increasing number of input frame, which is a model parameter, during training and inference. We have additionally conducted longer training input frames experiment. We added in our revisited paper the evaluation table for PSNR and LPIPS vs the number of frames during training & inference and the number of autoregressive steps. these results demonstrate that video duration can be extended while incurring only modest degradation in output quality. We also provide visualization of long video output on our project page.

---

### Official Review · Reviewer_ghJy · 2025-10-26

**Soundness:** 3
**Presentation:** 2
**Contribution:** 2
**Rating:** 4
**Confidence:** 4

**Summary:**

MoCtrl4D is a motion-promptable 4D Gaussian generation framework that takes a single image + user-specified motion trajectories as input and outputs dynamic 4D Gaussians in a single forward pass. Instead of text prompts, it encodes explicit 2D point trajectories into a “trajectory image” and feeds it directly into an existing L4GM video-to-4D reconstruction model without adding any new control modules. To improve motion fidelity, it proposes a new Vector Consistency Loss (VCL) to complement ARAP and better enforce rigid yet directionally correct deformation. Experiments on Objaverse show improved motion controllability (EPE) over L4GM while maintaining similar appearance quality.

**Strengths:**

1. Using explicit 2D trajectories as motion prompt is a very reasonable and underexplored alternative.

2. The UI in Fig. 4 is intuitive and gives users precise, fine-grained control over where to move and how to move.

3. The trajectory prompt is natively injected via “trajectory image” encoding, without increasing model size. This is clean and lightweight design.

4. VCL loss is a well-justified improvement over vanilla ARAP, addressing the translation vs. rotation ambiguity that ARAP alone cannot disambiguate.

**Weaknesses:**

1. The output 4D quality is still visually limited. Motions are generally small, conservative, and low-energy (mostly local deformations, not full articulated or large trajectory motion).

2. Appearance fidelity is mediocre. Textures look relatively flat/synthetic / Objaverse-style, clearly lagging behind 4Real, Imagen-3D, DynamiCrafter, or even modern feed-forward Gaussian works. It feels more like reconstructed animation than genuinely generated cinematic 4D content.

3. Relies almost entirely on synthetic Objaverse rigged assets, lacking evidence of generalization to real images or real natural motion.

4. Core novelty is not architectural. It is fundamentally an extension of L4GM with trajectory conditioning + new motion supervision loss. The “no extra control module” aspect is good engineering, but not necessarily a strong research novelty — many will see this as “just encoding trajectories into RGB channels” rather than a deep innovation.

**Questions:**

1. How robust is your method to natural photos or messy in-the-wild images?

2. Does your method still work if the motion trajectories are large or non-rigid?
Can it handle non-linear / high-amplitude / self-occluding motion? Or does it break because the model was only trained on Objaverse’s mild rig motions?

3. How do you guarantee the generated motion is physically plausible?
The VCL loss enforces relative vector consistency, but is there any failure case where Gaussians drift or jitter unnaturally? A failure case visualization would be important to assess reliability.

---

> ### Author Response · Authors · 2025-12-03
>
> We sincerely appreciate the reviewer’s valuable time, professional insights, and constructive questions. Your thoughtful feedback has greatly guided our revisions. Below are our detailed clarifications in response:
>
>
> $\textbf{Q1: How robust is your method to natural photos or messy in-the-wild images?}$
>
> MoCtrl4D currently specializes for artwork-style images. Since there is no readily available dataset for real world 4D assets, our model is trained only on objects. To be able to generalize to in-the-wild images requires careful self-supervised framework to train the model using in-the-wild data. We thank reviewer for the advice for further work.
>
>
> $\textbf{Q2: Does your method still work if the motion trajectories are large or non-rigid? Can it handle non-linear }$
>
> $\textbf{/ high-amplitude / self-occluding motion? Or does it break?}$
>
> According to the project page: https://moctrl4d.github.io/moctrl4d/, MoCtrl4D can handle non-linear / high-amplitude / self-occluding motion as shown in 4D generation section. Although ARAP and VCL are designed for rigid motions, prior work [1,2] shows that most real-world deformable motions can be decomposed into subparts that behave as if rigidly attached. Therefore, using these losses to guide nonrigid motion is theoretically well-justified.
>
>
> $\textbf{Q3: How do you guarantee the generated motion is physically plausible?}$
>
> Physical plausibility spans multiple levels, including (1) local continuity of object deformations, (2) consistency within rigid subparts, and (3) the more complex physical laws governing articulated or highly deformable objects, where decomposition into rigid subparts is less effective. Even though most real-world deformable motion can be decomposed into subparts that behave
> as if rigidly attached as stated in [1],[2], there are still the more complex cases beyond this assumption e.g. full elastic liquid-like models. While VCL does not aim to model all higher-level physical behaviors, it is designed to enforce the first two levels, thereby ensuring motion adheres to the most fundamental aspects of physical plausibility.
>
> [1]: Mosca: Dynamic gaussian fusion from casual videos via 4d motion scaffolds.
> [2]: Spatialtracker: Tracking any 2d pixels in 3d space

---

### Official Review · Reviewer_y7rG · 2025-10-30

**Soundness:** 2
**Presentation:** 2
**Contribution:** 2
**Rating:** 2
**Confidence:** 4

**Summary:**

The paper introduces a promptable 4D generation framework that enables motion control through user-defined point trajectories. It provides an intuitive interface for users to create motion prompts and animates static images interactively. Unlike prior methods that add complex modules, this approach integrates prompts directly into a base reconstruction mod. Additionally, it proposes a Vector Consistency Loss (VCL) to improve motion fidelity based on physical principles, overcoming the limitations of appearance-focused methods.

**Strengths:**

•  The paper addresses an important problem in 4D generation.

•  Providing users with an interactive experience is very interesting.

•  The results demonstrate a diverse range of cases.

**Weaknesses:**

1.	How does this work differ from SC4D [1], which also uses sparse point control?
[1] SC4D: Sparse-Controlled Video-to-4D Generation and Motion Transfer

2.	From the demonstrations, it seems that users need to perform multiple operations and drag several points. On average, how much time does it take to manipulate one example during inference? How many keyframes need to be adjusted — only the first and last frames, or multiple ones? Would using more frames improve accuracy while increasing interaction time?

3.	Previous works mainly focused on video-to-4D methods. With the recent progress in controllable video generation and stronger video base models (e.g., Wan, Veo3), how should we evaluate the quality of control signals? My understanding is that video-to-4D approaches use a short front-view video as input, while your method relies on a single image and sparse points. How do you compare the advantages of these two settings?

4.	Compared with rigging-based methods, where motion sequences are either learned from data or derived from motion libraries, your point-dragging approach seems to require heavier user interaction. For complex actions such as lifting a leg or dancing, rigging methods tend to produce smoother and more natural motion. In contrast, I feel that manually controlling motion through sparse points may lead to less natural and less continuous results.

5.	According to Table 1, there appears to be little performance improvement. Could this be because the model initializes L4GM weights and uses the same datasets? Compared with L4GM, this work employs the same architecture with an additional input signal, which makes the contribution seem somewhat incremental.

6.	Due to the limitations of the L4GM architecture, the model can only generate very short clips — the supplementary materials show results of just about one second. The visual results are not particularly impressive, and it’s unclear what practical value a one-second 4D generation result could have.

7.	In the supplementary materials, the ablation experiments (Exp1–5) are not clearly explained — it’s hard to tell what each experiment corresponds to.

**Questions:**

Please refer to the weeknesses.

---

> ### Author Response · Authors · 2025-12-03
>
> We are deeply grateful to the reviewer for the thoughtful comments and constructive questions. Our clarifications are provided as follows:
>
> $\textbf{Q1: How does this work differ from SC4D?}$
>
> SC4D is designed for video-to-4D generation and cannot use a single image with point prompts. The reason is that SC4D requires offline optimization of its MLP using L_ref and L_mask in eq (6) and (7) of the SC4D paper. Since L_ref requires RGB for each frame, it necessitates having a video as input for each new sample. In contrast, MoCtrl4D requires only a single static image per sample, eliminating the need for a full video to generate a 4D scene.
>
> $\textbf{Q2.1: How much time does it take to manipulate one example?}$
>
> The time depends on how detailed the user wishes the motion to be complicated. For simple actions, it only requires approximately 10 seconds. Please refer to UI demo in our project page: https://moctrl4d.github.io/moctrl4d/
>
> $\textbf{Q2.2: How many keyframes need to be adjusted?}$
>
> There are no keyframes required. The usage is dragging, and the speed of the mouse dragging and frame stride setting directly control the speed of the movement in the output 4D asset in one-to-one as correspondence manner.
>
> $\textbf{Q3: How do you compare the advantages of these two settings (video-to-4D methods vs single image and sparse points)?}$
>
> We view these settings as serving different purposes. While video-based 4D generation offers more comprehensive input, its applicability is limited by the requirement for pre-recorded footage. In contrast, the single-image and sparse-points setting is designed for users who have only a single image, enabling 4D generation without capturing a video while still maintaining high output quality.
>
> $\textbf{Q4.1: Point-dragging approach requires heavier user interaction than rigging-based methods?}$
>
> Rigging-based methods require keyframe manipulation in specialized 3D software, demanding skilled artists and often hours of iterative refinement for even simple, believable animations. In contrast, point-dragging is intuitive and requires no prior 3D animation training—users simply drag points once with a mouse and the smoothness of the animation is facilitated by MoCtrl4D. The entire point-dragging and 4D generation process takes under a minute.
>
> $\textbf{Q4.2: Manually controlling motion through sparse points may lead to less natural and less continuous results.}$
>
> Naturalness and smoothness are controlled by model inference and not by the input form. Since MoCtrl4D learns to generate natural motion given this kind of sparse points in training, it can do so in inference. Please refer to our project page for video evidence.
>
> $\textbf{Q5.1: Little performance improvement}$
>
> Although the overall quantitative metrics are comparable to the base L4GM framework, L4GM produces noticeably worse visual details by often predicts fading Gaussians or inconsistent colors in parts of the object. These issues have little impact on metrics but significantly affect the quality of the generated 4D assets. For example, on our project page in the 1st row of ablation section, the tail of the dragon disappearing or the arm of the character fading barely changes PSNR, yet is a serious visual flaw.
>
> $\textbf{Q5.2: The contribution seems incremental}$
>
> Our contributions are VCL, prompt injection without additional modules, and the novel motion-promptable 4D generation applicability.
> - VCL complements the shortcomings of the commonly used ARAP loss. While ARAP offers ambiguous and relatively weak guidance for 3D point motion, VCL provides explicit and unambiguous constraints as written in the paper.
> - Prompt injection without additional modules also reduces computational overhead and vRAM usage problems found widely in all available models. This new perspective may lead the field towards more resource-efficient practices.
> - The novel motion-promptable 4D generation task opens the unprecedented opportunity to reach wider use cases for users that only have a single image and wish to control the movement of that image.
>
> We believe the contributions are all reasonably significant and beneficial to the field.
>
> $\textbf{Q6: The model can only generate very short clips?}$
>
> We apologize for very short clips in the initial supplementary material. L4GM actually can generate videos up to 8 seconds during inference, making it practical in real use. We release the longer version on our project page: https://moctrl4d.github.io/moctrl4d/ and report additional experiments on frame length and output quality in the revised paper.
>
> $\textbf{Q7: In the supplementary materials, the ablation experiments (Exp1–5) are not clearly explained.}$
>
> We apologize for the confusion. The number of the ablation experiments in the supplementary materials are the same settings as in the main paper in Tab. Ablation. Each row corresponds to number 1 to 5 experiments, which we have improved the clarity of the table by adding number to each row.

---

### Author Response · Authors · 2025-12-03
**Discussion Summary**

Dear Reviewers, ACs, SACs. and PCs
Thank you sincerely for your precious time and unwavering dedication to this review process. We fully acknowledge that the workload has been particularly heavy this year, and we greatly appreciate your efforts amid such demands. To streamline the discussion, we have prepared a concise summary below. $\textbf{We hope it helps the ACs quickly grasp the key points of our rebuttal and facilitates a smoother, more efficient review process.}$

## Strength summary
We first sincerely thank the reviewers for their recognition of our work in 4 important aspects:

$\textbf{(1) Problem well addressed:}$ “important problem in 4D generation.” ($\textbf{y7rG}$), “very reasonable and underexplored” ($\textbf{ghJy}$), “important and underexplored” ($\textbf{RLzK}$).

$\textbf{(2) Interesting user Interaction:}$ “interesting interactive experience." ($\textbf{y7rG}$), “gives users precise, fine-grained control” ($\textbf{ghJy}$), “provides fine‑grained, expressive control beyond text/image prompts.” ($\textbf{Nryf}$).

$\textbf{(3) Clean model Design:}$ “clean and lightweight design” ($\textbf{ghJy}$), “improving temporal consistency” ($\textbf{Nryf}$).

$\textbf{(4) Novel loss function well design:}$ “well-justified improvement over vanilla ARAP” ($\textbf{ghJy}$), “specialised motion control” ($\textbf{Nryf}$).

## Concerns and Our Response
$\textbf{(1)	Comparisons with 4D generation methods:}$ We addressed this by providing video comparisons with L4GM, LASR and SP4D on our project page: https://moctrl4d.github.io/moctrl4d/. $\textbf{MoCtrl4D outperforms all the methods.}$

$\textbf{(2)	Results concerns:}$ Since we randomly selected samples in test set in the initial supplementary material, these samples indeed lack variability and amplitude of motions. We improved by selecting higher amplitude and motion variety to present with enough amount in our project page to ensure that the results are actually convincing. $\textbf{We also ensure that the generated motions are natural and smooth.}$ Please visit our project page: https://moctrl4d.github.io/moctrl4d/ for evidence.

$\textbf{(3)	Video duration concerns:}$ We conducted additional experiments for long motion and show visualization on our project page and ensure that $\textbf{MoCtrl4D can handle video length of practical use.}$

$\textbf{(4)	UI complexity concerns:}$ We recorded UI demo and showed on our project page to guarantee that $\textbf{UI is simple and easy for all users}$ without trained 3D skills.

$\textbf{(5)	Generalization to real-world objects concerns:}$ This is indeed limited by no readily available dataset for real world 4D assets. MoCtrl4D currently specializes only artwork-style objects. We thank reviewer for further work which may require careful self-supervision framework design.

$\textbf{(6)	Clearer explanation concerns:}$ We have addressed this issue by improving clarity of explanation as suggested in the revised paper.

$\textbf{(7)	Contribution concerns:}$ Our contributions are VCL, prompt injection without additional modules, and the novel motion-promptable 4D generation applicability.

-	VCL complements the shortcomings of the commonly used ARAP loss. While ARAP offers ambiguous and relatively weak guidance for 3D point motion, VCL provides explicit and unambiguous constraints as written in the paper.
-	Prompt injection without additional modules also reduces computational overhead and vRAM usage problems found widely in all available models. This new perspective may lead the field towards more resource-efficient practices.
-	The novel motion-promptable 4D generation task opens the unprecedented opportunity to reach wider use cases for users that only have a single image and wish to control the movement of that image.

We believe the $\textbf{contributions are all reasonably significant and beneficial to the field.}$


Finally, we sincerely thank the AC and reviewers for their invaluable constructive feedback, which has greatly refined our work. We hope this summary facilitates your final assessment of our paper.

---

### Meta-Review · Area_Chair_Wyp6 · 2026-01-10

**Summary:**

The paper was seen as a promising but incremental extension of L4GM, held back by limited generalization, weak visual results, and poor presentation.

**Reviewer Concerns:**

Addressed in the rebuttal

Missing baseline comparisons: authors added side-by-side videos versus SP4D, LASR and L4GM on a wider set of scenes.

Too few result videos: they released 30+ clips, including longer sequences (up to 8 s) and the same object under several motion prompts.

Disorganised supplementary material: an HTML project page was created with clear captions and grouped examples.

Need for a user study: a small 15-scene perceptual test was run and the numeric table was supplied.
Clarity of ablation settings: figure and table numbers were aligned and explained.

Still outstanding

Generalisation to real-world, “in-the-wild” images or non-rigid objects: training remains Objaverse-only; no new real-photo experiments were provided.

Limited motion amplitude and visual fidelity: although longer clips are shown, the largest displacements are still modest and texture quality is unchanged.

Incremental architectural novelty: the method is still an L4GM backbone plus trajectory-image input and VCL loss; no structural innovation was introduced.

Physical plausibility and failure modes: no systematic analysis of large, self-occluding or highly non-rigid motions, and no explicit failure cases demonstrated.

**Reviewer Scores:**

N.A.

---

### Decision · Program_Chairs · 2026-01-26

Reject